# Shared Molecular Pathways in Glaucoma and Other Neurodegenerative Diseases: Insights from RNA-Seq Analysis and miRNA Regulation for Promising Therapeutic Avenues

**DOI:** 10.3390/cells12172155

**Published:** 2023-08-28

**Authors:** Carlos Franciney Moreira Vasconcelos, Vinicius Toledo Ribas, Hilda Petrs-Silva

**Affiliations:** 1University of Medicine of Göttingen, 37075 Göttingen, Germany; 2Institute of Biophysics Carlos Chagas Filho, Universidade Federal do Rio de Janeiro, Rio de Janeiro 21941-902, Brazil; 3Institute of Biological Sciences, Universidade Federal de Minas Gerais (ICB/UFMG), Belo Horizonte 31270-901, Brazil; ribasvt@ufmg.br

**Keywords:** RNA-seq, neurodegeneration, glaucoma, miRNA regulation

## Abstract

Advances in RNA-sequencing technologies have led to the identification of molecular biomarkers for several diseases, including neurodegenerative diseases, such as Alzheimer’s, Parkinson’s, Huntington’s diseases and Amyotrophic Lateral Sclerosis. Despite the nature of glaucoma as a neurodegenerative disorder with several similarities with the other above-mentioned diseases, transcriptional data about this disease are still scarce. microRNAs are small molecules (~17–25 nucleotides) that have been found to be specifically expressed in the CNS as major components of the system regulating the development signatures of neurodegenerative diseases and the homeostasis of the brain. In this review, we sought to identify similarities between the functional mechanisms and the activated pathways of the most common neurodegenerative diseases, as well as to discuss how those mechanisms are regulated by miRNAs, using RNA-Seq as an approach to compare them. We also discuss therapeutically suitable applications for these disease hallmarks in clinical future studies.

## 1. Introduction

Neurodegenerative diseases, such as Alzheimer’s disease (AD), Parkinson’s disease (PD), Huntington’s disease (HD) and Amyotrophic Lateral Sclerosis (ALS) are characterized by a progressive loss of specific neuronal cell populations, in addition to other features such as abnormal protein aggregation, mitochondrial dysfunction, pro-oxidative and pro-inflammatory mechanisms [1,2,3]. Because of the neuronal cell loss, individuals affected by these types of pathology may present several neurological deficits, including cognitive, motor and behavioral dysfunctions, which impair their quality of life and ability to perform tasks [4,5]. AD and PD are the two most common neurodegenerative disorders in the elderly, which have been associated with comorbidities such as neuropsychiatric dysfunctions [6], ischemic episodes [7] and eye diseases, including Open-Angle Glaucoma (OAG) [8,9].

OAG is one of the most common eye diseases whose frequency increases with age, leading to vision impairment and, frequently, irreversible blindness [7,8,9,10]. Epidemiological predictions estimate that OAG will affect over 95.4 million individuals by 2030 [11]. The progressive visual loss and functional working effects, along with injuries affecting patient quality of life, have been reported as one of the most important public health problems, especially among elderly people [10,12]. Furthermore, OAG is characterized by the neurodegeneration of the retinal ganglion cells (RGCs), which are projection neurons located in the inner part of the retina that project their axons via the optic nerve to the superior responsive regions on the central nervous system (CNS), projecting a visual stimulus from the eye to the brain [10]. This neurodegenerative process leads to the stimulation of inflammatory and pro-oxidative mechanisms, as well as to mitochondrial dysfunction and progressive neuronal death, similar to the above-mentioned neurodegenerative diseases [8,13,14].

Currently, advances in RNA-sequencing (RNA-Seq) technologies have led to the identification of several molecular biomarkers, as well as to the accurate observation of global transcriptional alterations that are associated with the hallmarks of several diseases, including neurodegenerative ones [15]. Moreover, recent works have increased the identification of microRNAs (miRNAs) as a potential diagnostic biomarker in brain diseases [16,17]. miRNAs are small noncoding RNA molecules responsible for regulating protein homeostasis through binding to a targeted messenger RNA (mRNA), causing its cleavage, translational inhibition or degradation [15,17]. In fact, several miRNAs have been found to be specifically expressed in the CNS and involved in the regulation of the development signatures for neurodegenerative diseases, such as AD [15,18,19,20], PD [21], HD [22] and Creutzfeldt–Jacob disease (sCJD) [15], but data regarding transcriptional changes and miRNA regulation in glaucoma are still scarce.

In this review, we aimed to identify the similarities in functional mechanisms and activated pathways between the most common neurogenerative diseases, discuss how those mechanisms are regulated by miRNAs, using RNA-Seq as an approach to compare them, as well as highlight possible new therapeutical avenues for the application of miRNA-based drugs in future pre-clinical studies and clinical trials.

## 2. Glaucoma Similarity Hallmarks with Other Neurodegenerative Diseases

The retina is an extension of the brain having several similarities with it, such as specialized immune responses with local arrays of cytokines and surface molecules [23]. This is also exemplified in multiple well-defined neurodegenerative processes that happens in the brain and have manifestations in the retina (Figure 1). Additionally, ocular disfunctions frequently precede the diagnosis of CNS disorders [23,24]. Moreover, the retina can be analyzed by less invasive optical techniques than those used in brain examination, such as optical coherence tomography (OCT), and reflects pathological features in the brain, providing a window for an earlier diagnosis of CNS disorders, including neurodegenerative diseases [24,25].

In OAG patients, which is the most common form of glaucoma, there is a progressive increase in intraocular pressure (IOP) due to an impaired aqueous humor outflow [26]. However, recent research has shown that the sensitivity of the optic nerve to IOP is more relevant for glaucoma progression than the IOP itself [27]. In fact, some patients with increased IOP not always present neurodegeneration [26,28]. Additionally, the optic nerve head (ONH) is vulnerable to injury due to its over 1.5 million RGC unmyelinated axons, which can provide a unique environment for neurodegenerative processes in Glaucoma. There is a complex interplay between neuronal, glial, biochemical and vascular components in the ONH that can be influenced by alterations in the IOP [26,29,30]. These mechanisms can be impaired by aging but also by exacerbated inflammatory responses, such as astroglia activation and proliferation, increased deposition of collagen and changes in the extracellular matrix, which are not dependent on genetic inheritance or predisposition [26]. Moreover, it has been widely reported that immune-mediated events can trigger and contribute to the progression of several neurodegenerative diseases, including AD, PD, HD, ALS, among others [1,2,24,26].

The neuroinflammatory response is considered to be crucial for both regenerative and neurodegenerative mechanisms in the CNS, including the retina, and immunocompetent cells are key factors in the appearance of morphological changes, cell migration and proliferation related to these injuries [31,32]. Several studies demonstrated changes in the nerve fiber layer of the retina in patients with AD and PD, due to microvascular and structural alterations, as well as reduced blood flow associated with retinal thinning [24,33,34]. In patients with AD, neuroinflammation is a crucial hallmark of the pathology. The elevation in pro-inflammatory cytokines, due to a reactive astroglia response, leads to an accumulation of Aβ oligomers and tau protein plaques, resulting in neuronal loss [23,24]. Interestingly, the APP protein (precursor of Aβ oligomers) was found to be one of the most abundant proteins in the optic nerve and is transferred to the axon plasma membrane in small vesicles [23]. The elevation of the IOP has been reported to induce Aβ aggregate formation in RGCs in an experimental rat model of glaucoma [35]. Additionally, hyperphosphorylated tau protein was also detected in the retinas of glaucoma patients [7]. Similarly, PD patients demonstrated α-synuclein accumulation also in the retina, which underlies very early visual impairment symptoms [36]. In addition to the activation of the immune response, there is also evidence of transient optic disc microhemorrhages in patients regardless of the increased IOP levels, indicating an impairment of the blood–retina barrier (BRB) and the infiltration of circulating immune cells, which are associated with the progression of the disease [37,38,39]. This infiltration due the BRB rupture may represent some of the autoimmune components observed in humans and animal models [40]. Interestingly, the increased IOP can also cause an initial influx of neutrophils and macrophages that express, for example, oncomudulin and SDF1, stimulating axon outgrowth, showing a pro-regenerative mechanism under injury stress [41,42,43].

In the retina and optic nerve, the immune cells (e.g., microglia, astrocytes and Müller glia) act as a surveillance system, maintaining the homeostasis by releasing neuroprotective and growth factors, as well as in the clearance of cell debris [26]. Polyunsaturated fatty acids (PUFAs) are an important component of the cell membrane in the brain and are crucial for gene transcription, cell signaling and caspase activation [44,45]. RGC axons are myelinated distal to the lamina cribrosa. Cell culture studies showed that omega-3 PUFA can stimulate myelin production, and it is possible that a lack of omega-3 can lead to an alteration of myelin composition and to RGC synaptic disfunction [46]. PUFAs have a role in the treatment of PD [47], which shows, therefore, a correlation between the management of glaucoma and that of parkinsonism. Additionally, optic neuritis, which can be an early sign of multiple sclerosis (MS), is an autoimmune demyelinating and neurodegenerative disease. Its pathogenic mechanisms also include inflammation involving T- and B-lymphocytes and circulating innate immunity cells, as well as oxidative stress [48]. Patients with MS also show a reduction in the thickness of the retinal nerve fiber layer, as well as reduced optic nerve perfusion. OCT measures were able to detect RGC loss and axonal damage due to MS [48,49].

In humans, the inheritance of the *APOE4* allele increases the chance of developing AD [50]. Moreover, it was also reported that mouse models expressing *APOE4* demonstrated a higher microglia and astrocyte reactivity response, suggesting that even in the absence of AD, *APOE4* pro-inflammatory induction, as well as its correlation with dysfunctional microglial clearance of Aβ, may trigger neurodegeneration [50,51,52]. Additionally, *APOE4* showed to be protective against glaucoma and age-related macular disease (AMD), indicating, perhaps, that less reactive microglia are protective in the retina, whereas dysfunctional microglia are detrimental in AD [26,53]. The association of APOE with high-density lipoprotein (HDL) level was also reported as a neuroprotective target for clearance of α-synuclein in PD [54].

Other main hallmarks of neurodegenerative disorders are oxidative stress and mitochondrial dysfunction. The latter was reported in PD by the mediation of mitophagy and autosomal recessive phenotype induced by mutations in the PTEN-induced putative kinase 1 (PINK1) and PRKN (PARK2) genes [55]. PINK1 and LRRK2 are enriched in astrocytes, and their mutations can induce α-synuclein aggregation on the mitochondrial outer membrane [56,57]. In AD, on the other hand, Aβ aggregates cause an increase in the production of reactive oxygen species (ROS), affecting mitochondrial axonal transport, energy metabolism, mitophagy and mitochondrial fusion/fission [58]. Oxidative stress and mitochondrial dysfunction have also been associated with RGC degeneration, both in animal models and in humans [14,59]. In fact, aged mice have shown alterations in mitochondrial transport in RGCs and are more susceptible to an increased IOP than young mice [60]. The elevation in IOP have also been reported to affect mitochondrial bioenergetics in the brain’s visual cortex in rats, as well as lead to reduced ATP production, increased superoxide production and differences in the mitochondrial complex activity [61]. Moreover, RGCs and optic nerve fibers are rich in mitochondria, and the reduction in energy metabolism along with increased ROS production at the mitochondrial level are potential mechanisms to be considered in the etiopathogenesis of glaucoma [24]. Thus, the identification of cellular and molecular mechanisms related to RGC death is a major important step towards the discovery of new biomarkers and the development of therapeutic strategies against glaucoma.

## 3. RNA-Seq Analysis as a Tool to Search for Similar Pathway Regulation Mechanisms in Neurodegenerative Diseases

During the last decade, advances in RNA-Seq technologies have demonstrated the great potential of this approach to identify changes in gene networks and cellular pathways involved in various aspects of different diseases in a broad way [62]. Transcriptomic studies have provided analyses of expression variations of genes, as well as the quantification of their transcriptional level under different experimental conditions or in patients [63,64]. Moreover, access to these big data sets allows the interpretation and design of more effective functional analyses of candidate gene biomarkers, as well as of other transcripts [65], thus, providing an important tool for prospective investigations and future experimental validations.

With the emergence of Genome-Wide Association Studies (GWAS), researchers have been able to determine similarities between pathways and gene interaction networks across the entire genome in large cohorts, facilitating the identification of genetic variations associated with several neurodegenerative diseases. GWAS rely on significant top loci racks, and the assumption of genetic association similarities between diseases is made to identify overlaps; however, this assumption may overlook the fact that a strong genetic locus in one disease may not have a similar or significant impact in another disease [66]. Then, further investigations are necessary to discern each discovery’s unique genetic intricacies and avoid oversimplification in unraveling the complex genetic landscape that contributes to different conditions.

Previous work using RNA-Seq and gene expression profile meta-analyses showed interesting gene signature comparisons between neurodegenerative diseases. The analysis of different areas in 1270 post-mortem brain tissues from 13 patient cohorts affected by four diseases (AD, PD, HD and ALS) identified 243 examples of shared gene expression between the diseases, which were validated with another dataset comprising 205 samples from 15 cohorts [67]. Those shared genes were commonly reported in the literature of neurodegenerative altered pathways, including neuroinflammation, mitochondrial dysfunction, oxidative stress and synaptic plasticity impairments [66,68].

Although all neurodegenerative diseases have significant genetic and inherited components, whose contribution in AD, PD and ALS is estimated to be 60–80% [69,70], ~40% [71], and ~60% [72], respectively, most cases exhibit a complex etiology, involving sporadic elements [66]. Genetically based, inherited cases of these diseases have been attributed to rare mutations, such as those in the amyloid precursor protein (APP) and presenilin genes (PSEN1, PSEN2) for AD, α-synuclein (SNCA), *PRKN* (PARK2), PINK1, microtubule-associated protein tau (MAPT) and leucine-rich repeat kinase 2 or dardarin (LRRK2) for PD, cytosolic Cu/Zn superoxide dismutase (SOD1), alsin (ALS2), senataxin (SETX), TAR DNA-binding protein 43 (TARDBP) and synaptobrevin/VAMP (vesicle-associated membrane protein)-associated protein B (VAPB) for ALS [68,72,73]. Regarding glaucoma, although some elements of monogenic and polygenic Mendelian inheritance have been presented in several cases, common mutations in genes, such as MYOC and OPTN, or copy number variants in TBK1, only account for less than 5% of the genetic patterns of the disease [74]. Nonetheless, comparisons between public RNA-Seq data sets, also from other types of CNS neurodegeneration, identified common pathway alterations between different models of optical nerve damage, such as the DBA/2J mouse model, optic nerve crush, axotomy models [75]. These studies are in accordance with the enrichment of neuroinflammatory, pro-oxidative, and immune responses found for other neurodegenerative disorders. Furthermore, the expansion of these big data analyses using protein–protein interaction networks, KEGG pathways and gene ontology databases, could help researchers to find similar biomarkers that can further be used in pre-clinical and clinical studies. As an example, Li and collaborators [76] expanded a set of 10 common disease susceptibility genes shared between AD, HD and PD (e.g., ESR2, PARP1, GSK3B, UCHL1 and LRRK2) into a larger set of 1294 genes connected to the first set of genes in a protein network analysis. Furthermore, similar to mRNA expression profile analysis, several studies found that many important responsive mechanisms in neurodegenerative disorders are regulated by miRNAs. The up- and down-regulation of these molecules’ expression was described in several altered pathways in many neurodegenerative and brain disorders. Thus, their investigation could lead to the discovery of even more relevant biomarker candidates than mRNAs, due to their stability in many human biofluids and potential dysregulation in earlier stages of the diseases [68,77]. Adding that fact to the above-discussed sporadic and non-genetic character of many cases of neurodegenerative disorders, those molecules could bring new insights for future studies regarding the regulation of common mechanisms in brain homeostasis maintenance during neurogenerative injuries.

## 4. miRNAs as Potential Therapeutic Agents in Neurodegenerative Disorders

Several miRNAs were found to be expressed specifically in the CNS and to be dysregulated in neurodegenerative diseases [18,19,20,21,22]. These dysregulated miRNAs can target key genes involved in disease-associated pathways, thereby influencing disease progression. For example, miR-29a was shown to regulate genes involved in extracellular matrix remodeling, which is implicated in the pathogenesis of AD and PD [19,21]. miR-132 and miR-212 were found to be downregulated in AD and HD, and their targets include genes involved in synaptic plasticity and neuroprotection [18,22]. miR-124, which is highly expressed in the brain, was shown to regulate neuronal differentiation and is dysregulated in various neurodegenerative diseases [15]. These examples highlight the importance of miRNAs in modulating disease-associated pathways and provide potential targets for therapeutic interventions.

However, given the similarities in the functional mechanisms and activated pathways between glaucoma and other neurodegenerative diseases, it is plausible to hypothesize that miRNAs may also play a role in glaucoma pathogenesis. RNA-Seq analysis can be a valuable tool to identify differentially expressed miRNAs in glaucoma and explore their potential involvement in disease-related pathways. By comparing the transcriptomic profiles of glaucoma samples with those of other samples of neurodegenerative diseases, common dysregulated miRNAs and pathways can be identified, providing insights into the shared molecular mechanisms underlying these diseases. Furthermore, identifying miRNAs as potential diagnostic biomarkers in neurodegenerative diseases opens the possibility of developing miRNA-based therapeutics. Modulating the expression or activity of disease-associated miRNAs may offer a novel approach to the treatment of glaucoma and other neurodegenerative diseases. Strategies such as miRNA replacement therapy or miRNA inhibition using antisense oligonucleotides (ASOs) can be explored.

The biogenesis of miRNAs involves a series of steps. It begins in the cell nucleus, where miRNAs are transcribed from specific genomic loci by RNA polymerase II or, in some cases, RNA polymerase III [78,79]. These primary transcripts, called pri-miRNAs, can be several hundred to thousands of nucleotides long and often contain a stem–loop structure. The next step involves the processing of pri-miRNAs by an enzyme complex known as the microprocessor complex, which consists of the RNase III enzyme Drosha and its co-factor DGCR8 (DiGeorge syndrome critical region 8) in humans [78,79]. The microprocessor complex cleaves the pri-miRNA into a smaller hairpin-shaped RNA molecule known as the precursor miRNA (pre-miRNA). The pre-miRNA is then exported from the nucleus to the cytoplasm by the nuclear export protein Exportin-5 [78,79]. In the cytoplasm, the pre-miRNA is further processed by another RNase III enzyme called Dicer, along with its co-factor TRBP (TAR RNA-binding protein) or PACT (protein activator of the interferon-induced protein kinase) in humans [78,79]. Dicer cleaves the pre-miRNA, resulting in the formation of a small RNA duplex. The RNA duplex consists of two strands: the mature miRNA (guide strand) and the miRNA* (star strand). The miRNA* strand is typically degraded, while the mature miRNA strand is loaded into a multi-protein complex known as the RNA-induced silencing complex (RISC) [78,79]. Within the RISC complex, the mature miRNA guides the complex to target mRNAs with complementary sequences in their 3′ UTRs.

The mechanism of action of miRNAs involves base-pairing interactions between the miRNA and its target mRNA. The miRNA recognizes its target mRNA through partial complementarity between the miRNA sequence and the target mRNA sequence. The binding of the miRNA to the mRNA can lead to translational repression or mRNA degradation, depending on the degree of complementarity and other factors. In the case of translational repression, the binding of the miRNA to the mRNA blocks the translation process, preventing the synthesis of the corresponding protein [78,79]. Overall, miRNAs play a critical role in regulating gene expression by modulating the stability and translation of target mRNAs. They are involved in a wide range of biological processes, including development, cell differentiation, proliferation, and response to environmental cues. The dysregulation of miRNAs has been implicated in various diseases, such as cancer, neurodegenerative disorders and cardiovascular diseases, highlighting their significance in both normal physiology and pathology. In fact, several potential therapeutic miRNAs have been identified as promising targets for the treatment of different neurodegenerative disorders. Here are some examples.

1. Alzheimer’s disease (AD): AD is characterized by the accumulation of amyloid-beta (Aβ) plaques and neurofibrillary tangles in the brain. miRNAs such as miR-124, miR-29a and miR-34a have been implicated in AD pathogenesis. These miRNAs regulate the expression of genes involved in Aβ production, tau phosphorylation and neuroinflammation [80]. miR-124 and miR-29a/b regulate the Aβ synthesis pathway by targeting the 3′ UTR of the *BACE1* mRNA precursor, promoting the regulation of the APP expression [81,82]. miR-124 increased the number of apoptotic and necrotic cells in vitro in a PC12 cellular AD model and participated in the regulation of *BACE1* expression [81]. miR-34a regulated Tau expression in vitro, both in neuroblastoma and in HEK 293 cell models [83,84] and, additionally, enhanced NMDA and AMPA receptors expression in the APP/PS1 mice model [84]. Modulating the levels of these miRNAs could potentially mitigate AD-related pathology.

2. Parkinson’s disease (PD): PD is characterized by the loss of dopaminergic neurons in the substantia nigra of the brain and α-synuclein neurofilaments’ accumulation. miRNAs such as miR-134a, miR-133b, miR-7, miR-153 and miR-34b/c were found to play a role in PD pathogenesis. These miRNAs regulate the expression of genes involved in neuronal survival, dopamine synthesis and mitochondrial function [85,86]. The reduction of miR-133b enhanced the overexpression of *SNCA*, which induced the production of α-synuclein, leading to a major sensitivity of dopaminergic neurons to oxidative stress [87]. The activation of PI3K/Akt signaling can also induce α-synuclein overexpression, while the activity of miR-7 and miR-153 can inhibit the PI3K/Akt signaling pathway. Moreover, the PI3K/Akt, LRRK2/ERK1/2 and PI3K/PTEN signaling pathways are involved in neuronal apoptosis, oxidative stress and neuroinflammation. miR-34a and miR-153 were associated with the regulation of those mechanisms, reducing the progression the disease [86,88]. Thus, modulating these miRNAs may have therapeutic potential in PD.

3. Huntington’s disease (HD): HD is caused by a mutation in the huntingtin gene, resulting in the accumulation of a toxic mutant huntingtin protein in neurons. miRNAs such as miR-9, miR-124, miR-140 and miR-34a were identified as potential therapeutic targets for HD. These miRNAs modulate the expression of genes involved in neuronal survival, mutant huntingtin protein clearance and neuroinflammation. miR-9 increased the expression of *RE1* in leukocytes and was downregulated in peripheral leukocytes of HD patients [89]. miR-140 regulated disintegrin and metalloproteinase 10 (*ADAM 10*) expression, which reduced the excessive cleavage of the synaptic protein N-cadherin at the postsynaptic densities, impairing synaptic transmission [90]. miR-124 is a crucial regulator of neuronal differentiation in neurodegenerative processes. The injection of miR-124 increased the expression of the neuroprotective proteins PGC-1α and BDNF in a mice model and downregulated the SRY-related HMG box transcriptional factor 9, a repressor of cell differentiation [91]. Modulating the levels of these miRNAs could potentially attenuate HD pathology.

4. Amyotrophic lateral sclerosis (ALS): ALS is characterized by the progressive degeneration of motor neurons in the spinal cord and brainstem. miRNAs such as miR-34a, miR-183, miR-218 and miR-335 have been implicated in ALS pathogenesis. These miRNAs regulate the expression of genes involved in motor neuron development, survival and neuroinflammation. For instance, miR-34a acts by targeting the 3′UTR of the *XIAP* mRNA precursor, promoting protective effects against oxidative stress-induced apoptosis through *SIRT1* [5,92]. miR-335, miR-183 and miR-218 expression is downregulated in ALS patient’s serum and in mouse models; this affects several disease-linked mechanisms, such as the regulation of caspase 7 expression [93]. Modulating the levels of these miRNAs could potentially modulate the immune response and reduce oxidative stress and apoptosis in ALS.

5. Open-angle Glaucoma (OAG): OAG is characterized by an increase in the IOP and a consequent progressive degeneration of RGCs, causing blindness in most cases. miRNA such as miR-204, miR-124, miR-125a and miR-125b have been implicated in OAG pathogenesis. Among the large number of miRNAs involved in the pathophysiology of the retina, miR-204 appears to be one of the major players in retinal development and disease [94]. Additionally, the involvement of miR-124, miR-125a, miR-125b and miR-204 in the development of retinal cells in adult mice has already been confirmed [95]. miR-125b was shown to play indispensable functions during the differentiation and maintenance of retinal pigment epithelium (RPE) cells [96]. miR-29 was demonstrated to be a negative modulator of the expression of collagens and other key components of the extracellular matrix in human TM cells and to decrease cytotoxicity in the presence of chronic oxidative stress [97].

It is important to note that while the modulation of miRNAs shows promise as a therapeutic strategy for neurodegenerative diseases (Table 1), further research is needed to fully understand the roles and potential clinical applications of miRNAs. Additionally, developing safe and efficient delivery methods for miRNA-based therapies remains a challenge in the field. Nonetheless, targeting specific miRNAs holds significant potential for the development of novel treatments for several diseases. While neurodegenerative diseases are distinct disorders with unique pathological mechanisms, there is some overlap in the miRNAs that have shown therapeutic effects across different neurodegenerative diseases. Here are a few examples of miRNAs that demonstrated potential therapeutic benefits in multiple neurodegenerative diseases:

1. miR-124: miR-124 is one of the most abundant miRNAs in the brain and is involved in regulating neuronal development and function. It was shown to have therapeutic effects in AD, PD, HD and OAG. In AD, miR-124 was found to target genes involved in Aβ production and tau hyperphosphorylation [81,94]. In PD, it was shown to protect dopaminergic neurons and regulate inflammation [98,99]. In HD, miR-124 was reported to modulate the expression of genes associated with mutant huntingtin protein clearance and neuroinflammation [91,100]. In OAG, miR-124 was reported as a key regulator of the development and maintenance of retinal cells in adult mice [96].

2. miR-9: miR-9 is involved in neuronal development and has been implicated in several neurodegenerative diseases, including AD, PD and HD. In AD, miR-9 was found to target genes associated with Aβ production and neuroinflammation [101]. In PD, it was shown to regulate genes involved in dopaminergic neuron survival and mitochondrial function [16,98]. In HD, miR-9 was reported to modulate genes related to mutant huntingtin protein clearance and neuroinflammation [99].

3. miR-146a: miR-146a is known for its role in regulating immune responses and inflammation. It has been implicated in AD, PD and multiple sclerosis (MS). In AD, miR-146a was shown to regulate genes involved in neuroinflammation and Aβ clearance [102]. In PD, it was found to modulate inflammation and dopaminergic neuron survival [103]. In MS, miR-146a was reported to regulate immune cell activation and the inflammatory response [104].

4. miR-155: miR-155 is involved in immune system regulation and has been implicated in AD, PD and MS. In AD, miR-155 was found to target genes involved in Aβ production and neuroinflammation [105]. In PD, it was shown to regulate inflammation and dopaminergic neuron survival [106,107,108]. In MS, miR-155 was reported to modulate immune cell activation and the inflammatory response [109,110].

**Table 1 cells-12-02155-t001:** miRNAs involved in the regulation of neuroprotective mechanisms.

Disease	miRNA	Targeted Genes	Physiological Process	Reference
**Alzheimer’s disease**	miR-124	*BACE1*	Reduction of Aβ oligomers production and neuroprotection	[81]
	miR-34b	*VAMP2*, *SYT1*, *BCL2*	Regulation of presynaptic activity and anti-apoptotic mechanisms	[111]
	miR-30a	*BDNF*	Regulation of synaptic plasticity and neuroprotective mechanisms	[112,113]
	miR-34c	*P53*, *SIRT1*	Neuronal development	
	miR-125b	*BACE1*	Attenuated Aβ toxicity in Aβ-treated N2a cells via targeting BACE1	[114]
	miR-132	No reports	Prevents the apoptosis of neurons and is involved in the regulation of synaptic plasticity, learning and memory, as well as in reducing tau hyperphosphorylation	[19,115,116]
	miR-124	No reports	Regulates the hyperphosphorylation of the tau protein in cell cultures via the PI3K/AKT/GSK-3 pathway	[94]
	miR-29a miR-29b	*BACE1*	Regulation of APP and beta-site APP-cleaving enzyme 1 (BACE1)	[117]
	hsa-miR-483 hsa-miR-486	*MAPK1/3*	Direct repression of Erk1/2 and reduction of tau phosphorylation	[118,119]
	miR-135a miR-384	No reports	Repression of BACE-1 and APP-cleavage	[120]
	miR-342	*ANK3*	Regulation of neurogenesis and neuroprotection in an AD mouse model	[121]
	miR-20a	*BCL2*, *MEF2D*, *MAP3K12*	Regulation of gene expression during brain development.	[122]
	miR-329	No reports	Regulation of activity-dependent dendritic outgrowth of hippocampal neurons	
	MiR-200a	*BACE1*, *PRKACB*	Reduces Aβ accumulation and tau hyperphosphorylation, respectively	[123]
**Parkinson’s disease**	miR-150	*AKT3*	Downregulation of the proinflammatory cytokines IL-1β, IL-6 and TNF-α	[114]
	miR-124	No reports	Attenuates microglia activation, improves survival of dopaminergic neurons and reduces α-synuclein aggregation	[98,99]
	miR-132 miR-92a miR-27a miR-148a	*GLRX*	Regulates the activation and loss of microglial cells	[111,124]
	miR-146a	No reports	Regulation of anti-inflammatory mechanisms	[104]
	miR-153 miR-214 miR-34b/c miR-7	No reports	Downregulation of α-synuclein expression, preventing its neurotoxicity	[125,126,127,128]
	miR-26b	*TAK1*, *TAB3*	Suppressing neuroinflammation by downregulating the activators of NF-Kβ	[129]
	miR-34a	*D1*, *SIRT1*, *BCL-2*	Neuronal development, brain ageing metabolic regulation p53/miR-34a/SIRT1 pathway	[114]
	miR-29	No reports	Regulates T helper 1 (Th1) differentiation and targets the transcription factors T-bet and Eomes, resulting in the repression of IFN-γ production	[130]
	mir-29b	*BAD*, *BIM*, *BID*, *BIK/NBK*, *BNIP3*, *BLK*, *HRK*, *NOXA*, *and EGL-1*	Regulates neuronal survival by targeting genes in the pro-apoptotic BH3-only family to inhibit apoptosis	[131]
	miR-30e	*NLRP3*	Regulates neuroinflammation by reducing the inflammatory cytokines TNF-α, COX-2 and iNOS	[132]
**Amyotrophic Lateral Sclerosis**	miR-128	*THY1*	p53 Pro-apoptotic pathway regulation	[133,134]
	miR-191	*BDNF*	Neuronal and immune cell apoptosis regulation	[133]
	miR-24	*BIM*, *PUMA*	Thy1/ Thy2 balance regulation	
	miR-27a miR-155 miR-142	No reports	Regulation and control of oxidative stress	[135,136]
	miR-300	*VASH2*	Neuron differentiation	[137]
	miR-450b	*SOX2*, *PTPRZ1*	Neuron differentiation and neurogenesis regulation	
	miR-34a	*XIAP*	Protective against oxidative stress-induced apoptosis through SIRT1	[5,92]
**Huntington’s disease**	miR-124a	No reports	Regulator of neuronal differentiation and morphology	[100,138]
	miR-128	*HTT*	Neuronal survival, metabolic pathways, particularly cholesterol (affected by mutant HTT)	[139]
	miR-122	*AACS*, *ADAM10*, *BCL2*		
	miR-132	*ITPKB*	Neuronal development and survival	[111,140]
	miR-9	*HTT*, *CoREST*	Increases the expression of RE1 in leukocytes	[89]
	miR-140	*ADAM10*	Synaptic function regulation	[90]
	miR-34a	*D1*, *SIRT1*, *BCL-2*	Neuronal development and brain ageing metabolic regulation p53/miR-34a/SIRT1 pathway	[141]
**Glaucoma**	miR-92a	*KALRN*	Axonal guidance signaling, Ephrin B signaling, synaptogenesis signaling pathway	[142]
	miR-99b	No reports	Regulation of autophagy, senescence, neuroinflammation	[143]
	miR-125b	*LIN28B*	Adhesion, tight junctions and TGF-β signaling	[141]
	miR-192 miR-10a miR-10b	No reports	Neurogenesis, aging, apoptosis and autophagy	[144]
	miR-375	*BDNF*		
	miR-143	*LMO4*	Regulation of autophagy, apoptosis, senescence, neuroinflammation	[145]
	miR-221	No reports	TGF-β and neurotrophin signaling	[146]
	miR-451a	No reports	Adhesion, tight junctions and TGF-β signaling	[147]
	miR-486	*LMCM*, *LMX1B*, *PTPN1*	TGF-β signaling regulation	[146]
		*TXNRD2*	Antioxidant action of vitamin C, mitochondrial dysfunction, thioredoxin pathway, vitamin C transport	
	miR-124 miR-204	No reports	Development and maintenance of retinal cells in adult mice.	[95]
	miR-29	*p53*	TGF-β signaling regulation and antioxidant effects.	[97]

These are just a few examples of miRNAs that have shown therapeutic potential in multiple neurodegenerative diseases (Figure 2). The overlapping effects of these miRNAs suggest common underlying mechanisms or pathways involved in the pathogenesis of different neurodegenerative diseases. However, it is important to note that the specific targets and mechanisms of action of these miRNAs may vary in different diseases, and further research is needed to elucidate their precise roles and potential for therapeutic interventions.

## 5. Challenges and Potential Solutions for miRNA Therapy

miRNA-based therapy has become a promising strategy for the treatment of a variety of diseases. Nonetheless, the effects of miRNAs on multiple targets, the development of novel approaches based on their identification, the successful development of delivery methods and of chemical formulations to overcome the crossing of BBB and the description of their off-target effects are still a challenge for translating pre-clinical studies into clinical applications [148,149]. Several studies have already demonstrated possible alterations in signaling pathways in relation to the finding of miRNA target genes in widespread RNA-Seq data analysis, but additional investigations are needed to further develop miRNA-based therapies based on miRNA overexpression, functional inhibition or knockdown to control neurodegenerative injuries.

One of the main problems that need to be addressed while using miRNA-based therapies is to avoid miRNA degradation by RNases. Chemical modifications could solve this issue, but the short half-life of some constructs is still a challenge [148]. Another issue that needs to be considered is the time course of the miRNA effects. For example, the application of miR-124 increased neuronal survival in the ischemic brain, whereas a late administration did not show the same effect [149]. Some miRNAs’ “off-target” effects could also activate unwanted pathways that counteract their protective effects [149,150]. An example of the effects of miRNAs at multiple levels is provided by the miR-15-miR-16 cluster, which acts by downregulating multiple targeted apoptotic proteins, including BCL-2 and MLC1, affecting the same cancer hallmarks [151,152]. Despite these challenges in miRNA-based therapeutics, cell receptor- and tissue-specific targeted AAV have been demonstrated in numerous studies using preclinical models and even in current clinical trials for gene therapy, providing a promising way to deliver miRNA to specific sites [153]. These multiple target capacities of some miRNAs may also potentially boost the therapeutic effects compared to other RNA-based therapies, such as synthetic those using siRNAs or ASOs that act on target-specific genes.

Among the possibilities to overcome the challenges of using miRNA-based therapy, we can name (1) the use of ‘tiny’ antisense RNAs, which are short sequences of 7–8 nucleotides that target the 5′-seed of miRNAs and are able to inhibit an entire miRNA family sharing the same seed sequence [149]. Although the use of shorter sequences could increase the possible off-target effects, it was already demonstrated that targeting two miR-122 family members using locked nucleic acid (LNA)-modified antimiRs in vitro led to few off-target effects, with no measurable effects on the mRNA-targeted complementary sequence [154]; (2) applying the metronomic miRNA therapy, which is defined as a frequent administration of limited drug doses over a prolonged period, aiming to achieve a low active dose range with no excessive toxicity or immunological reactivity [149]; (3) combinatorial RNA therapy, which can also reduce the use of the required doses; for example, the application of miR-155 therapy was able to reestablish cancer cell sensitivity to chemotherapy [155], and, similarly, miR-34a therapy sensitized lung cells to radiotherapy [156]; (4) the use of small-molecule inhibitors of miRNAs (SMIRs), which can act on miRNAs at the transcriptional and post-transcriptional levels or even influence their processing to regulate specific miRNAs [157].

These are just a few examples of the application of miRNA therapy in combination with other approaches in order to overcome the current challenges. There is still an urge to better understand the multiple targeting actions of miRNAs and their effects, but the development of designed chemical formulations and new delivery methods, as well as the identification of new miRNA targets and the reduction of their off-target effects can lead to future successful clinical trials and effective miRNA-based treatments for several disorders, including neurodegenerative diseases.

## 6. Conclusions

RNA-Seq technologies provide a powerful tool for researchers to evaluate a wide volume of data that could lead to the identification of new biomarkers for numerous diseases, as well as of pathway alterations, protein network alterations and physiological events, including the regulation mediated by miRNAs, that are common between neurodegenerative disorders. These small molecules play a crucial role in diverse biological processes by regulating mRNA molecules, leading to mRNA degradation or translational repression. In the context of neurodegenerative diseases, miRNAs have emerged as promising therapeutic targets, with specific miRNAs showing dysregulation and implication in disease pathogenesis and progression. Modulating the expression or activity of these miRNAs holds great potential for novel therapeutic strategies. Some miRNAs, like miR-124, miR-9, miR-146a and miR-155, have demonstrated therapeutic effects in multiple neurodegenerative diseases, including Alzheimer’s, Parkinson’s, and Huntington’s diseases. These miRNAs target genes associated with key pathological processes, indicating shared molecular mechanisms among different neurodegenerative diseases. However, it is crucial to recognize that these miRNAs’ specific targets and mechanisms of action may vary across diseases, highlighting the complexity of miRNA-mediated regulation. While miRNAs represent a promising avenue for therapeutic intervention, more research and clinical trials are needed to fully explore their potential and translate these findings into effective treatments for neurodegenerative disorders, including neurodegenerative eye diseases like open-angle glaucoma.

## Figures and Tables

**Figure 1 cells-12-02155-f001:**
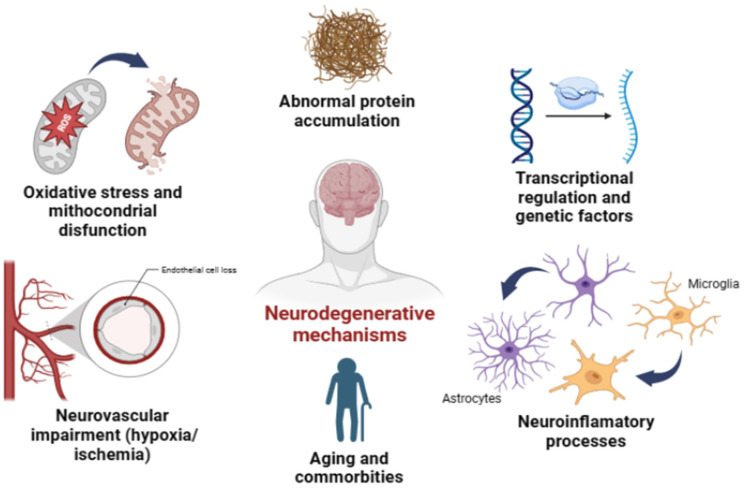
Similar neurodegenerative mechanisms between glaucoma and other neurodegenerative diseases. It has been described that several mechanisms are common between the above-mentioned neurodegenerative diseases, such as neuroinflammatory mechanisms that are associated with a major activation of astrocytes and microglia and/or by circulating immune cell infiltration due to neurovascular impairments. Genetic factors could also be implied to a minor degree and with differences between the diseases, but with some common mechanisms. Mitochondrial dysfunction and impaired energy metabolism along with increased ROS production are also major common mechanisms the lead to neuronal loss. Also, we observed that the presence of protein aggregates (e.g., Aβ oligomers, α-synuclein and hyperphosphorylated tau protein) can be a common feature of AD, PD and glaucoma. All these factors are aggravated with aging and in most cases are accompanied by other disturbs. Figure made using BioRender (Toronto, ON, Canada).

**Figure 2 cells-12-02155-f002:**
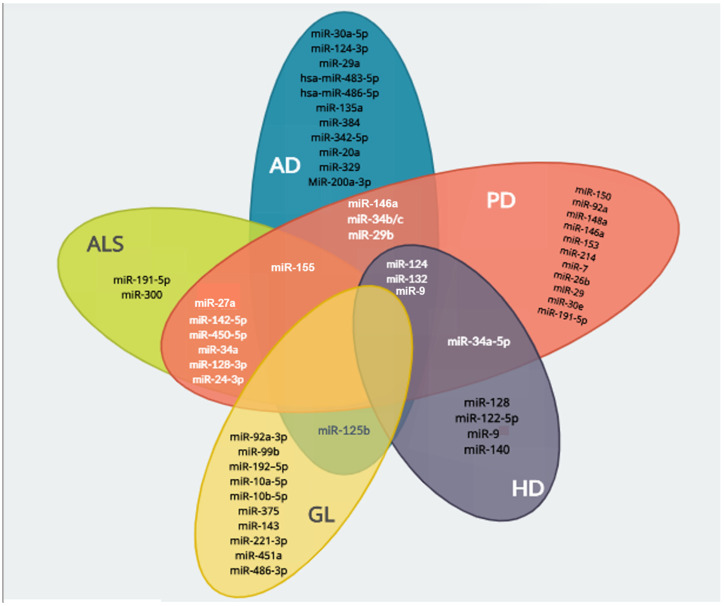
Venn’s diagram showing the overlapping miRNAs described for the different neurodegenerative diseases. The diagram demonstrates an overlapping of some miRNAs between the above-mentioned diseases. AD and PD share miR-34b/c, miR-29b, miR-146a and miR-155; PD and ALS share miR-27a, miR-155, miR-142-5p, miR-450-5p, miR-34a miR-128-3p and miR-24-3p; PD and HD share miR-34a-5p; AD, PD and HD share miR-124 and miR-132; AD and glaucoma (GL) share miR-29, miR124 and 125b. There is no overlapping of the neuroprotective miRNAs described in the present review that might share a similar biological function in all the discussed neurodegenerative diseases.

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
