# Peer review of "Shared Molecular Pathways in Glaucoma and Other Neurodegenerative Diseases: Insights from RNA-Seq Analysis and miRNA Regulation for Promising Therapeutic Avenues"

_cells, 2023, doi:10.3390/cells12172155_

Round 1

Reviewer 1 Report

I would recommend to include data obtained by other methods (RT-PCR, etc.), if such data are available.

Author Response

We would like to thank all reviewers for their contributions to this study. We made the necessary corrections in the points raised in the text, references and / or figures, here are the answers to the questions raised:

Reviewer 1:

1.I would recommend to include data obtained by other methods (RT-PCR, etc.), if such data are available.

ANSWER:

Thank you for your suggestion. We think it would be an interesting addition to the final text, however, in this review we sought to focus on big data analysis. Therefore, we believe that add data from other methods would change the focus of this review, which aims to analyze the shared molecular pathways in glaucoma and other neurodegenerative diseases based on RNA-Seq and the potential of using miRNAs as therapeutic tools.

Reviewer 2 Report

Dear authors, 

This is a great review analyzing the shared molecular pathways in glaucoma and other neurodegenerative diseases based on RNA-Seq and the potential of using miRNAs as therapeutic tools. The paper is well-organized and documented. I only have a few comments: 

1. Apart from the potential of miRNAs as therapeutic tools, based on the RNA-Seq findings, can you refer to other therapeutic agents which could help on these diseases (i.e. antioxidants regulating ROS pathways,..). I think it is also important to mention other approaches, although the focus of the review is miRNA. 

2. Can you expand a little bit more section 5? Maybe refer to other diseases where also, they are assessing the possibility of using miRNA as a therapeutic tool. 

Thank you very much. 

Author Response

We would like to thank all reviewers for their contributions to this study. We made the necessary corrections in the points raised in the text, references and / or figures, here are the answers to the questions raised:

Reviewer 2:

  1. Apart from the potential of miRNAs as therapeutic tools, based on the RNA-Seq findings, can you refer to other therapeutic agents which could help on these diseases (i.e. antioxidants regulating ROS pathways,..). I think it is also important to mention other approaches, although the focus of the review is miRNA.

ANSWER:

We would like to thank you very much for the suggestion. We did mention other approaches, as you may find in the sections 4 and 5. However, in this review we sought to focus on big data analysis approaches, as well as other types of RNA-based therapies. Lines: 257-261 (section 4), lines (436-439, 441-455),

2.Can you expand a little bit more section 5? Maybe refer to other diseases where also, they are assessing the possibility of using miRNA as a therapeutic tool. 

ANSWER:

Thank you for the suggestion. We agree that this could be a great addition for the section, and you can find this addressed in the revised version of the manuscript. In addition, by your response, I think you mentioned section 4, instead of 5, where we talk about the therapeutic actions of some miRNAs in different diseases already reported, if I got wrong, please let me know.

Reviewer 3 Report

This paper provides valuable insights into the potential shared molecular pathways and the exploration of miRNA regulation among neurodegenerative diseases, including Glaucoma. However, to enhance the abstract's readability and impact, some clarity and additional details are needed. Here are some comments and suggestions for further improvement:

Line 249-250: The authors state, "In the context of glaucoma, there is still a lack of data regarding the transcriptional changes and miRNA regulation specific to this disease." This statement is not entirely accurate, as several miRNAs have been found to be dysregulated in Glaucoma and have been reported as potential diagnostic biomarkers or therapeutic targets. For instance, miR-29b, miR-200c, miR-204, miR-24, miR-143-3p, miR-125b-5p, and miR-1260b have all been implicated in Glaucoma. The authors should include a mention of these findings in the abstract and possibly add them to Table 1, which presents miRNAs related to Glaucoma.

Line 264-307: The detailed description of miRNA seems somewhat redundant. It might be better to move this part to the introduction section, where it can provide background information on miRNA and its relevance to neurodegenerative diseases and Glaucoma. This would help streamline the abstract and improve its focus on the study's specific objectives and findings.

Author Response

We would like to thank all reviewers for their contributions to this study. We made the necessary corrections in the points raised in the text, references and / or figures, here are the answers to the questions raised:

Reviewer 3:

1.Line 249-250: The authors state, "In the context of glaucoma, there is still a lack of data regarding the transcriptional changes and miRNA regulation specific to this disease." This statement is not entirely accurate, as several miRNAs have been found to be dysregulated in Glaucoma and have been reported as potential diagnostic biomarkers or therapeutic targets. For instance, miR-29b, miR-200c, miR-204, miR-24, miR-143-3p, miR-125b-5p, and miR-1260b have all been implicated in Glaucoma. The authors should include a mention of these findings in the abstract and possibly add them to Table 1, which presents miRNAs related to Glaucoma.

ANSWER:

We would like to thank you for the observation and suggested additions. We have now corrected this in the abstract and in the text, as well as included information about miRNAs and glaucoma in the revised version of the manuscript. Additionally, the first phrase of the paragraph: "In the context of glaucoma, there is still a lack of data regarding the transcriptional changes and miRNA regulation specific to this disease." Was deleted from the reviewed version.

2.Line 264-307: The detailed description of miRNA seems somewhat redundant. It might be better to move this part to the introduction section, where it can provide background information on miRNA and its relevance to neurodegenerative diseases and Glaucoma. This would help streamline the abstract and improve its focus on the study's specific objectives and findings.

ANSWER:

We would like to thank you for the observation and suggested additions. Nonetheless, as the introduction needs to be more objective, as usually is seen in other papers of Cells Journal, we have simplified that specific mentioned part regarding the biogenesis and action mechanism to make the text clearer and straightforward.